# Patient-Derived Xenotransplant of CNS Neoplasms in Zebrafish: A Systematic Review

**DOI:** 10.3390/cells11071204

**Published:** 2022-04-02

**Authors:** Beatriz E. Sarmiento, Santiago Callegari, Kemel A. Ghotme, Veronica Akle

**Affiliations:** 1School of Medicine, Universidad de Los Andes, Bogotá 11711, Colombia; beatrizelenasarmiento@gmail.com (B.E.S.); s.callegari@uniandes.edu.co (S.C.); 2Department of Neurosurgery, Fundación Santa Fe de Bogotá, Bogotá 111071, Colombia; kemelgg@unisabana.edu.co; 3Translational Neuroscience Research Lab, Faculty of Medicine, Universidad de La Sabana, Chía 250001, Colombia

**Keywords:** patient-derived, xenotransplant, zebrafish, glioblastoma, neuroblastoma, CNS neoplasms

## Abstract

Glioblastoma and neuroblastoma are the most common central nervous system malignant tumors in adult and pediatric populations. Both are associated with poor survival. These tumors are highly heterogeneous, having complex interactions among different cells within the tumor and with the tumor microenvironment. One of the main challenges in the neuro-oncology field is achieving optimal conditions to evaluate a tumor’s molecular genotype and phenotype. In this respect, the zebrafish biological model is becoming an excellent alternative for studying carcinogenic processes and discovering new treatments. This review aimed to describe the results of xenotransplantation of patient-derived CNS tumors in zebrafish models. The reviewed studies show that it is possible to maintain glioblastoma and neuroblastoma primary cell cultures and transplant the cells into zebrafish embryos. The zebrafish is a suitable biological model for understanding tumor progression and the effects of different treatments. This model offers new perspectives in providing personalized care and improving outcomes for patients living with central nervous system tumors.

## 1. Introduction

### 1.1. Neoplasms of the Central Nervous System 

Worldwide, 308,102 patients were diagnosed with neoplasms of the central nervous system (CNS) in 2020, causing an estimated 251,329 deaths [1] The World Health Organization (WHO) classifies brain tumors ranging from a genuinely benign tumor (grade I), in which the complete tumor resection can be curative, to a high-grade malignant tumor (grade IV) for which, even with combined treatment, the expected survival is only approximately 22–24 months [2]. Between 2013 and 2017, the annual prevalence of CNS tumors in the United States was 6.4 per 100,000 persons, with an estimated mortality of 4.3 per 100,000 persons [3]. Eighty-five to 90% of the CNS tumors affect the brain [4], the most frequent being anaplastic astrocytomas and glioblastomas (GB), which account for 38% of primary brain tumors, followed by meningiomas and pituitary tumors [4].

Brain tumors are the most common type of solid childhood cancer, second only to leukemia as a cause of pediatric malignancies [5]. The incidence ranges from 0.3–2.9 per 100,000 live births [6,7]. Similar to their adult counterparts, the most common histological type of tumors are gliomas (of which the most common is pilocytic astrocytoma) and embryonal tumors (of which the most common is medulloblastoma) [8]. When considering infants, the most common extracranial solid malignant tumor is neuroblastoma (NB), and it is the most common cancer overall in infants younger than one year, with an incidence rate of 65 per million infants [9,10].

One of the most studied CNS neoplasms is GB [11] since it is the most common primary malignant brain tumor in adults and has one of the highest mortality rates of any neoplasm. The reported GB survival rate is 3.3% at two years and 1.3% at three years [12]. Currently, the standard of care is complete surgical resection combined with radiotherapy and temozolomide [13]. However, recurrences are common [12]. After numerous clinical trials evaluating different treatment strategies, most of them failed to add significant results in improving patient outcomes [14]. 

### 1.2. Main Pathways in the Development of Neoplasms of the CNS 

Numerous investigations have studied the origin and pathways involved in developing CNS neoplasms [15], with controversies regarding the tumor-initiating cells that undergo mutations to proliferate into actual tumors [16]. The neural progenitor cells, also called neural stem cells (NSC), consists of oligodendrocyte precursor cells and astrocytes. They can proliferate if they receive a specific pathologic insult [15]. Nevertheless, there is no conclusive evidence that the NSCs are necessary or exclusive players in the formation of gliomas [17].

Due to the high heterogeneity of these neoplasms, defining a common pathway is challenging [15]. The Cancer Genome Atlas Project conducted in 2008 showed that 80% of the GB analyzed had altered signaling of the tyrosine kinase receptor, p53, and retinoblastoma protein (RB) [18]. Based on these classifications, Verhaak et al. in 2010 proposed four genomic subtypes, namely mesenchymal, classic, proneural, and neural [19]. One additional subdivision classifies GB in primary (de novo) or secondary (progressing from WHO grades I and II to grade IV) [20].

There is agreement that the critical pathways in the tumorigenesis of these neoplasms are the p53 pathway (p53/MDM2/4/p14ARF), the PTEN/NF1/RTK pathway (EGFR/RAS/NF1/PTEN/PI3K), and the RB pathway (p16INK4a/CDK4/RB) [15]. Mutations in TP53 and Nf1 appear in various grades of astrocytomas and enable them to evade apoptosis [21,22,23]. The PTEN pathway involves receptor tyrosine kinases (EGFR, PDGFR) and their associated pathways, which enable cell growth [24]. PTEN and NF1 modulate cell cycle entry in NSCs [25,26]. Various mutations in these proteins enable high-grade malignant glioma driven by MYC oncoprotein and highly penetrant GB [27,28]. Finally, RB regulates the G1/S checkpoint in the cellular cycle, causing increased mitotic activity [29,30].

### 1.3. Xenotransplantation in Zebrafish 

In cancer research, xenotransplantation is the transfer of human cancer cells into a different species [31]. It is considered a human-in animal disease model with unique advantages and challenges compared to other models [31]. Standard cancer xenotransplant models include mammals, such as mice, rats, rabbits, dogs, and monkeys, due to their genetic and physiologic similarities with humans [32,33,34,35,36]. However, the most common xenotransplantation of human cancer cells employed in preclinical research occurs in immunocompromised mice. Most of the great discoveries in preclinical research in cancer are thanks to mice models due to their multiple benefits, such as their homology to human physiology, but mainly due to the development and maintenance of specific strains through genetic manipulations and careful breeding under laboratory conditions, such as immunodeficient strains, through many years of research with the model. The model, though very advantageous, also has limitations.

The zebrafish (*Danio rerio*) model has emerged as a meaningful biological model to study cancer due to its genetic, molecular, and histological similarities with humans [32]. It has numerous characteristics that make zebrafish a suitable candidate for xenotransplant, and ingeniously complement and enrich cancer research. Embryos are easy to obtain, breed, and manipulate, with a daily production of hundreds, which facilitates extensive studies and high-throughput screening on in vivo assays [37]. Hundreds of embryos/larvae can be maintained on multi-well plates at a low cost [37,38,39]. Embryos develop rapidly, and in just 48 h, their nervous system is functional, reacting to stimuli through reflexive motility [40]. Zebrafish embryos and larvae are optically transparent, making them useful for dynamic live fluorescent imaging. This feature is particularly advantageous as tumor progression and some cellular processes are readily evident through microscopy techniques [41,42]. 

In zebrafish, adaptive and innate immune systems are highly similar to mice and humans [43]. Adaptive immunity is functional and morphologically mature between the second to the third week after fertilization, while innate immunity starts to appear on the first day after fertilization [44,45,46,47,48]. Thus, a lack of mature immune response during early larval development facilitates the transplantation of numerous cell types into zebrafish [37,49,50,51]. Although the zebrafish model has several benefits, there are some limitations to this model. Zebrafish larvae thrive in optimal conditions at 28 °C [52], which is below the temperature for proliferation and survival of mammalian cells (37 °C) [53]. Thus, most protocols maintain the larvae at an intermediate temperature of 32–33 °C with no significant consequences. Xenotransplantation can be performed in dozens of adult zebrafish and more than a hundred zebrafish larvae in a single day by a single operator, facilitating high throughput screening in cell transplantation studies [54,55,56,57,58,59,60,61]. Notwithstanding, the technique has many challenges and requires skill and many training hours for a high rate of injection and survival. Despite all the benefits, cancer xenotransplant studies in zebrafish are not numerous.

The first study of transplantation of human cells into zebrafish was published by Lee et al., 2005 [62]. This group injected human melanoma cell lines into the blastula stage of zebrafish embryos [62]. Since then, xenotransplantation using cancer cell lines into zebrafish larvae has been helpful to evaluate multiple diseases (for example, liver cancer [63,64,65], pancreatic cancer [66], colon cancer [49], ovarian carcinomas [67], gliomas [68,69], glioblastoma [70], and breast cancer [71,72], among many other types of cancer).

On the other hand, xenotransplant in zebrafish larvae from patient-derived samples is less common. Four years after the xenotransplantation of the first melanoma cell line in 2009, Marques et al. transplanted small, labeled samples from patients with pancreatic, colon, and stomach adenocarcinomas into the yolk sac of zebrafish [73]. Since then, many other types of primary cells have been transplanted (leukemia [74,75,76], breast cancer [77,78], pancreatic ductal adenocarcinoma [79], melanoma cells [80], gastric cancer cells [81], neuroendocrine tumor cells [82], and colorectal cancer cells [49]. The research in this field continues to grow with xenotransplantation of different types of cancer from both cell lines and patient-derived samples [36].

### 1.4. Xenotransplantation of Primary Tumors of the Central Nervous System (CNS)

The zebrafish model is a novel tool for exploring essential aspects of GB and NB [83]. In this model, implantation of human glioma cells yields similar morphology and characteristics to those reported in mice [84]. Opposed to extensive studies evaluating xenotransplants of human CNS neoplasms cell lines, studies on patient-derived primary xenotransplant are scarce. This review aimed to thoroughly describe the results of xenotransplantation of patient-derived CNS tumors in zebrafish models.

## 2. Methods

We followed the Preferred Reporting Items for Systematic Reviews and Meta-Analyses (PRISMA) statement’s guidelines and recommendations [85] to reliably structure the information synthesized in this review.

### 2.1. Eligibility Criteria

We considered studies using primary xenotransplant of central nervous system neoplasms in zebrafish, published between 2005 and 2021, without language restriction. We excluded studies whose primary goal was not patient-derived xenotransplants.

### 2.2. Search Strategy

We systematically searched PubMed, Scopus, Google Scholar, Scielo, Lilacs, and ProQuest. We searched for each subtype individually based on the WHO classification of CNS tumors. We utilized only the histologic classification and avoided terms describing explicit genomic mutations to maintain a broad scope and include studies published before the standardization of reporting CNS neoplasms according to the new WHO classification.

The search syntax for the databases consisted of:[CNS tumor name] AND [zebrafish OR Danio rerio] AND [transplant OR xenotransplant OR microinjection];[CNS tumor name] AND [zebrafish OR Danio rerio] AND [patient-derived] AND [transplant OR xenotransplant OR microinjection] (Figure 1).

### 2.3. Selection Process

Two authors performed the primary article screening individually. Each one reviewed the title and abstract of the selected studies. Then, we classified the listed articles in a three-tier classification system: selected, discarded, or unsure. Studies classified as “selected” or “discarded” and concordant in both lists were reviewed for the study or eliminated, respectively. The authors reviewed those classified as unsure in full text and discussed the findings. Finally, all authors reviewed the selected articles to ensure proper classification in the study. We solved any discordance regarding the classification by consensus.

### 2.4. Data Management

After selecting the articles, we extracted and collected data in an excel spreadsheet. The team analyzed and discussed the findings. We resolved disagreements by consensus. 

### 2.5. Data Items

We extracted the following data form each article: title, abstract, authors, publication year, type of tumor used, demographic data of the patients used for the sample, the role of the primary xenotransplant in the study, the method for stabilizing the sample, zebrafish model used (embryo, larvae, adult), site of xenotransplant, comparison between primary and cell culture transplant within the study, laboratory methods used to assess the transplant, screening of compounds and which ones used, molecular pathways studied, assessment of proliferation, angiogenesis, apoptosis, metastasis, and dissemination within the animal.

### 2.6. Data Synthesis

After collecting the information, we established categories with similar characteristics across studies. The main categories analyzed were tumor classification, cell pathways, zebrafish model used, site of xenotransplant, cell management method used, comparison between established cell lines, and screening of compounds. Finally, we reported our results according to the Preferred Reporting Items for Systematic Reviews and Meta-Analyses (PRISMA) guidelines.

## 3. Results

Of the 439 studies identified with the initial search, 5 studies presented results of patient-derived solid CNS tumor samples that were xenotransplanted into the zebrafish model. Glioblastoma (GB) and neuroblastoma (NB) were the xenotransplanted brain tumors with a clear predominance of GB. We classified and contrasted the findings from these five studies (Table 1) in the following categories: primary culture conditions, xenotransplant model, pathways studied, assessment of tumor viability and progression, and drug screening.

### 3.1. Primary Culture Conditions

#### 3.1.1. The Sample

The reviewed studies used patient-derived pediatric and adult high-grade GB [86,87,88,89] and NB tumor cells [90]. High-grade brain tumors are a heterogeneous sample that contains cancer stem cells, also called tumor propagating cells, “bulk cells,” and cells of varying differentiation status, which differ significantly from cell lines (homogeneous sample) [86]. In this aspect, the researchers of this article characterized the tumor sample in terms of methylation pattern, copy number alterations, DNA mutations, and stem cell markers to compare with primary culture patterns. They found that all the tumors strongly correlated with the GB primary culture, where the characteristic present in the tumors was preserved and maintained during the maintenance of the cells in culture [86]. Despite the similarities between tumor samples and GB primary cultures, the authors indicated an inter-patient heterogeneity of the tumors [86].

#### 3.1.2. Cell Culturing

GB primary cultures can grow in two different ways: adherent conditions and tumorspheres [86,87,88,89]. GB tumor samples were chemically dissociated, cultured in media for stem cells with serum-free conditions, supplemented by different growth factors, and maintained at 37 °C with 5% CO_2_ [86,88,89]. However, Rampazzo in 2013 established the GB primary culture in tumorspheres with a single variation, placing the cell culture in an atmosphere of 1.5% oxygen, 5% carbon dioxide, and using the balance nitrogen technique [87].

Common supplements used for maintaining GB cell cultures include B27 without vitamin A, epidermal growth factor (EGF), basic fibroblast growth factor (FGF), and penicillin/streptomycin (1%) [86,88,89]. Under these conditions, GB cells cultures accurately mirrored the tumors they arose from [86]. The cells were proliferative, positive for stem cell markers (OLIG2, nestin, SOX2, and Vimentin), and without apparent alterations in morphology or growth rate [86]. These cells could respond to differentiation cues and were stable during prolonged culture [86]. Meanwhile, Rampazzo and co-workers maintained the GB primary culture for no more than two consecutive passages to avoid long-term culture-related effects [87].

The specific growth factors utilized in cell cultures are highly relevant. Wenger and co-workers noted that GB cells cultured with only EGF proliferated faster than cells cultured with FGF-2 [86]. The EGFR-amplified GB cells retained amplification for at least five but lost it after 15 passages. Conversely, culturing the cells in FGF-2 helped retain the amplification at passage 15 [86]. Culturing the cells in only EGF or EGF + FGF- 2 did not change the methylation profiles. According to Wegner, culturing cells with serum-free conditions is essential to avoid cellular differentiation, vital to GB primary culture [86].

NB primary cultures were established in tumorspheres (short-term cultures) and cultured in mostly the same conditions as GB primary culture [90].

### 3.2. Xenotransplant Model 

Microinjection is possible in zebrafish at different developmental stages of development (embryo, larvae, and adult) and distinct locations (vitelline duct, pericardium, brain ventricles, among others) (Figure 2). The xenotransplant model used in these five publications using glioblastoma cells (GB) varied among studies, though most of them (4/5) aimed for realistic models injecting in brain regions. Wenger and colleagues injected cells into the brain of 48 h postfertilization (hpf) wild-type strain larvae [86]. Banasavadi-Siddegowda selected larvae with a clear midbrain-hindbrain boundary (approximately 16–24 hpf larvae) and injected 50 cells in that specific location [89]. Rampazzo and colleagues used wild-type and transgenic zebrafish larvae at seven days post fertilization (168 hpf) and used 100–150 cells per shot and did a double injection at the periventricular zone [87]. Finally, Pudelko and colleagues used zebrafish embryos at the blastula stage from five different zebrafish strains and injected 100 DiI labeled cells into the cell mass of the blastula [88].

Wrobel and colleagues used a similar approach for the xenotransplant of NB cells [90]. They used the E4/6 wild-type strain for the volume experiments and Tg (fli1:EGFP) line for the dissemination assessments [90]. For the first one, they injected 140–250 CM-DiI labeled cells into the yolk sac of 48 hpf larvae; for the dissemination assessments, they used the same type of larvae but injected in the perivitelline cells and reduced the number of cells (100 cells) [90] (Table 1).

### 3.3. Pathways Studied 

A few cellular pathways have been the focus in primary xenotransplants studies. Banasavadi-Siddegowda assessed the protein arginine methyltransferase 5 (PRMT5) pathway [89], which is a part of the PRMT family and plays a crucial role in histone modification (methylation) and, therefore, gene expression [91]. Increased expression of PRMT5 is common in high-grade gliomas, with a negative correlation with patient survival [92]. Therefore, it is one of the main therapeutic targets in several studies. The action of PRMT5 inhibitors resulted in apoptosis of differentiated GB cells and drove the GB patient-derived cells into a non-replicative senescent state in the in vivo model [89].

On the other hand, Rampazzo explored the Wnt pathway [87]. Forcing differentiation of neural stem cancer cells could also be a therapeutic approach, and the Wnt pathway has been suggested as one of the physiological mediators to regulate differentiation of normal neural progenitor cells [87]. Rampazzo assessed the role of the Wnt pathway in multiple ways, emphasizing how the expression of the molecule from the transplant receiver affected the expression of the human GB cells [87].

They used two types of transgenic zebrafish: one with a Wnt reporter mutation, which suggested a xenotransplant zone of high expression of Wnt in the midbrain-hindbrain boundary, and the other with a conditionally suppressed Wnt signaling, via overexpression of DKK [87]. For the first one, beta-catenin expression was significantly increased in GB cells, suggesting activation of the Wnt pathway in the patient-derived GB cells [87].

When characterizing the phenotypic expression of these cells, there was a progressive loss of *nestin* and increased expression of *b-III-tubulin* and *MAP2* (indicating that the zebrafish brain induced a phenotypic shift of transplanted GB cells towards neuronal fate) [87]. When using the second transgenic (the one with a suppressed Wnt pathway), GB cells maintain the expression of *nestin*, *b-III-tubulin*, and *Ki67* markers [87].

Figure 3 represents the numerous pathways involved in the development and establishment of glioblastoma (GB) at the molecular level. The Wnt pathway is highlighted, which is the only one partly studied in GB patient-derived cells, so far. The immense opportunity window for future research exploring the remaining pathways is evident from this figure. 

### 3.4. Assessment of Tumor Viability and Progression

Transplantation of patient-derived GB and NB cells into zebrafish allowed evaluating tumor progression in terms of cell proliferation, angiogenesis, metastatic potential, and apoptosis (Table 1).

#### 3.4.1. Proliferation

The proliferation of patient-derived GB cells was evaluated in vitro by Wenger et al., 2017 and Banasavadi-Siddegowda et al., 2017 using flow cytometric analysis [86,89]. Meanwhile, Rampazzo and colleagues in 2013 and Pudelko et al., 2018 evaluated them in vivo using live confocal microscopy, flow cytometric analysis, time-lapse confocal imaging, and light-sheet microscopy, respectively [87,88]. In this scenario, GB’s primary culture (with the culture conditions explained earlier) proliferates successfully and can maintain the primary culture viable for a time frame (>30 passages in culture) [86,89].

Rampazzo and co-workers monitored the proliferation of patient-derived GB primary culture in vivo into the brain of zebrafish [87]. They showed small cells with round morphology at four hours post-injection (hpi), while at 48 hpi, GB cells increased in size and exhibited cellular projections followed by axonal and neurite outgrowth [87]. Moreover, they found that GB cells treated with Wnt3a had a significant increase in P21cip1, indicating cell cycle arrest and proliferation inhibition [87]. Pudelko and colleagues, using tumor volume measurements and immunohistochemical analysis, showed that primary GB cells proliferate in zebrafish embryos [88].

Wrobel and co-workers showed that NB cells were mitotically active following transplantation, and these pediatric tumor cells survive and proliferate at rates such as those observed in patient tumors [90]. Besides, by evaluating the potential propagation into the perivitelline space, they showed numerous disseminating tumor cells in the tail region at 72 hpi [90]. Together, these experiments indicate that viable human tumor cells are present in the tail region of zebrafish larvae following the engraftment into the perivitelline space [90]. The combination therapy evaluated here promotes tumor regression, possibly inducing pro-apoptotic pathways [90].

#### 3.4.2. Angiogenesis

Pudelko et al. observed active tumor angiogenesis after transplanting primary GB cells into Tg(fli:EGFP) strains of zebrafish embryos with a fluorescent blood vessel system [88]. By using light-sheet real-time imaging, they confirmed an active tumor vascularization in embryos transplanted with GB#18 cells that expressed a red fluorescent protein [88]. Using molecular techniques, Rampazzo and co-workers showed a dramatic decrease in the key mediators c-JUN, VEGF, LDHA, GAPDH, and ALDOA, indicating a robust decrease in proliferation, angiogenesis, and glycolysis related genes [87].

#### 3.4.3. Metastatic Ability/Migration

Primary GB cells transplanted into the blastula stage embryos migrated from their injection site into the CNS and established a congregated, orthotopic tumor within 24 h after transplantation, indicating a general migration capacity of GB cells [88]. After apical injection, 74% of tumors grew in the fore/midbrain, while 25% grew in the tail, with no evidence of tumor growth in the hindbrain. On the other hand, after basal injection into blastula embryos, 73% of tumors grew in the fore/midbrain, 3% in the hindbrain, and 25% in the tail [88]. The migration observed was not influenced by the transplantation site into the blastoderm [88].

#### 3.4.4. Apoptosis

Apoptosis in patient-derived GB cells is a helpful molecular marker for evaluating effective treatments in vitro. Banasavadi-Siddegowda and co-workers showed the capacity of the inhibitor CMP5 to increase apoptotic cells in GBMDC significantly [89]. At the same time, GBMNS decreases the self-renewal capacity of GBMNS without affecting viability, i.e., CMP5 Drives GBMNS Toward Senescence [89].

### 3.5. Drug Screening

Temozolomide is one of the main therapeutic agents used in the clinical treatment of glioblastoma. It has been tested on GB primary culture in vitro [86] and zebrafish xenotransplanted model in vivo [88]. The GB primary culture (GB#18) established by Pudelko and co-workers (2018) did not respond to temozolomide, leading them to test different tyrosine kinase inhibitors (TKI) utilized in different clinical trials [88]. Erlotinib and the other TKI had limited but significant effects on the primary GB#18 transplants [88]. Other therapeutic agents used in vitro for GB primary cultures include Etoposide, Vincristine, and Temozolomide. The response to the drug treatment varied considerably between the cultures (BPC-A7 and BPC-B0) [86], which could suggest differential effects to treatments among patients.

Banasavadi-Siddegowda et al. tested for the first time PRMT5 inhibitors on GB primary cultures in both conditions, in vitro and in vivo [89]. Among the different PRMT5 inhibitors evaluated, CMP5 blocks the PRMT5 activity, inducing apoptosis of differentiated cells and senescence in immature primary tumor cells [89].

Doxorubicin is one of the most commonly used chemotherapeutics in neuroblastoma treatment [93]. Wrobel and co-workers tested Doxorubicin, Vorinostat, Panobinostat, and Tubastatin in zebrafish in vivo avatars. They found out that combination therapy involving Vorinostat and Doxorubicin substantially attenuated the progression of primary tumors in zebrafish larvae and showed decreased tumor cell dissemination [90].

## 4. Discussion

To the best of our knowledge, this is the most comprehensive systematic review describing in depth the results of xenotransplants of patient-derived GB and NB cells into a zebrafish model. As a biological cancer model, zebrafish has essential characteristics that have been widely reviewed [36,52,94,95,96]. Zebrafish cancer avatars with thoroughly described protocols help study patient-derived tumor cells [97]. However, the publications that use this model for the xenotransplantation of CNS tumor samples are scarce, probably since patient-derived tumor samples are not easy to obtain or since maintaining the primary cells in culture is challenging.

In this review, we highlight critical aspects related to brain tumor progression, treatment, and challenges of the model. Our review can be helpful for a better approach in using patient-derived CNS tumor cells with zebrafish as an in vivo model. The limitations mainly seen in the literature for the use of this model are the sample availability, the success of establishing the primary culture, and the cells’ ability to retain their molecular characteristics from the original tumor sample.

### 4.1. Heterogeneity

Brain Tumor samples from GB are usually very heterogeneous: a mixture of cancer and stem cells- GSCs with complex interactions between them and among different cells within and surrounding the tumor [19,98,99]. This feature represents a unique challenge to investigators and one of the primary considerations of the model. In the case of GB, most cells cannot recapitulate a phenocopy of the original tumor, and only a small subpopulation (GSCs) can form a tumor by themselves [100,101]. This heterogeneity not only can alter results at the laboratory but in the clinical practice, in which GSCs are notorious for their resistance to conventional chemotherapeutic agents and are the source for tumor initiation as well as recurrence [102]. For example, one of the factors that determine the effectiveness of temozolomide at the individual level is the tumor heterogeneity of the patient [88]. For these reasons, one of the main challenges in studying GB is to develop optimal culture conditions that preserve the molecular heterogeneity of the original tumor. 

The articles addressed this problem with different methods. The authors mentioned here demonstrated that it is possible to obtain and establish a primary culture for an extended period and maintain the molecular characteristics of the cells from the original sample [86]. Although there are slight methodological differences in methods to culture cells between the articles, there is a consensus on the use of growth factors and the non-use of fetal bovine serum to avoid cell differentiation from the original tumor sample. Different authors support this choice, stating that the patient-derived GSCs exposed to serum eventually lose the ability to form a tumor in vivo [101,103,104,105]. However, other authors used to establish primary cell culture of human a solution of 10% of Fetal Bovine Serum [106,107]. Meanwhile, if the objective was to establish GB neurospheres cultures from the primary culture of fresh tumor sample, the authors did not use a bovine serum [105].

This review showed the scarcity of therapeutic compounds evaluated on GB primary cultures and the limited number of pathways studied. Perhaps this scenario is due to the limitations of having access to the patients’ samples. For this reason, most of the studies of primary xenotransplant do not study several known pathways involved in GB. Nevertheless, this will likely change in the search for new compounds targeting CNS neoplasms. Hence, establishing methodological referents is crucial for optimal and comparable results.

Due to the high heterogeneity, it is unlikely that a single compound will work with the same effectiveness for every patient. Therefore, this scenario makes it necessary to search for new therapeutic agents specific to CNS tumors and pathways in specific patients. A thoroughly described mechanism of action in the pathways could be part of an array of new therapies oriented to personalized medicine. Thus, establishing xenotransplantation of cells derived from patient brain tumors is a reliable and valuable platform with great potential for personalized therapy, similar to T-cell acute lymphoblastic leukemia [74]. We considered that zebrafish as a biological in vivo model could be the best option for personalized medicine due to the numerous advantages described previously. Just to mention a successful application, a study testing FOLFOX 5-fluorouracil, oxaliplatin, folinic acid in zebrafish xenotransplant of patients’ colorectal cancer tumor cells found that the response of different samples to treatment correlated with clinical results (52). 

### 4.2. Primary Xenotransplant vs. Cell Lines

Most of the biological characteristics such as proliferation, apoptosis, and drug screening, among others, have been evaluated in vitro using cell lines [70,107,108,109,110,111,112]. Although they have been instrumental in understanding tumor behavior, the mere fact of being a cell line does not represent the actual heterogeneity of a tumor sample from a patient. For example, Allen and co-workers using short tandem repeat analysis and mitochondrial DNA, compared the original U87MGB cell line with the same cell line distributed from the American Type Culture Collection (ATCC) and cell line service (CLS) [113]. They discovered that the original U87MG did not genetically match the U87MG provided by ATCC and CLS; even though the cells are of CNS origin, these cells did not match the reference line of origin [113]. 

From this point of view, the zebrafish xenotransplant offers the possibility to study main biological aspects such as tumor proliferation, angiogenesis, and metastasis in vivo, while keeping the unique molecular characteristics of the tumor cells. The fundamental importance of evaluating these biological aspects in vivo is that it is possible to evaluate the proliferation pattern (size, cellular projection) of the individual tumors of each patient; as observed for NB patient-derived tumor samples, in which the neuroblastoma cells were mitotically active following implantation. The cells survive and proliferate at rates similar to those inside the patient [90]. 

In their most recent review, Tucker and co-workers emphasized the importance of patient-derived neuroblastoma xenografts to validate tumor progression, molecular targets, and drug resistance [114]. Even though murine systems are highly valuable and effective models, they have limitations such as low engraftment rates, long latency to tumor formation, and the high cost of initiating and maintaining experiments over extended periods [114]. In addition, it is typical to transplant patient-derived tumor cells in murine models after the third cell passage or later, while in zebrafish, it could be carried out directly after cell dissociation [115,116]. Table 2 presents the main differences between zebrafish and murine models using xenotransplants.

Wrobel et al. (2020) described a rapid preclinical zebrafish xenotransplant neuroblastoma model for the first time, where tumors grew in a brief period and allowed rapid therapeutic intervention [90]. At the moment, it is the only work published with patient-derived neuroblastoma xenotransplant into zebrafish. We included this study to analyze the effectiveness of this platform for this type of brain tumor.

Numerous studies have shown that tumor cell xenotransplantation of different origins in zebrafish allows the analysis of different cancerous events such as invasion, metastasis [62,73,117], angiogenesis [50,51,118,119], and cancer therapies [50,51,118,119]. In this way, the xenotransplant of the human cells into zebrafish can address many stages of carcinogenesis and also shows that human cells communicate effectively with recipient fish tissue [120]. Assessing distinct stages of carcinogenesis might be more challenging in murine models since it requires more invasive methods such as biopsies or postmortem analysis, whereas, in zebrafish larvae, distinct features can be appraised under confocal and light-sheet microscopy in living individuals.

Vittori and co-workers 2015 described the methodologies of xenotransplantation studies on zebrafish involving tumor progression, angiogenesis, and screening of potential therapeutic agents [52]. Additionally, they highlight the importance of the injection site into zebrafish for evaluating the carcinogenesis process from different glioma cell lines [52]. However, the xenotransplant model using patient-derived GB cells was not standardized and varied widely from studies. 

The main limitations of all of the studies were the small number of available patient-derived brain tumor samples and the challenge of culturing the cells under conditions that maintain the tumor’s properties. According to our analysis, the way to overcome these issues is to use growth factors (EGF, FGF, and B27) and cultivate stem cells with tumor proliferative capacity in serum-free media [86,87,88,89,90]. These are the optimal conditions required for a primary culture establishment with the same original tumor characteristics, as far as possible. Although, the composition of the medium can vary between laboratories, with no agreement yet on the best protocol to use.

This review analyzed different variables regarding the microinjections, including injection site, hours post fertilization of the zebrafish, number of cells injected, and zebrafish with specific mutations. The site of injection is one of the most relevant variables to consider. For example, injecting the cells in the vitelline duct, as described by Wrobel et al., 2020 [90] will bring the cells into a nutrient-rich environment where they can thrive and proliferate, but it would not be as helpful in evaluating crossing the blood-brain barrier. Additionally, injections in this area form microtumors that thrive for four to five days before regressing [70]. Changes in cell expression that limit the reproducibility of the results from the experiment to clinical care are one of the main challenges patient-derived xenotransplants face [118]. Therefore, the microinjection performed directly at the CNS could be one of the possible ways to reduce these variables. The studies in this review show that GB can be successfully injected into encephalic regions (ventricles, mid-hindbrain boundary, and blastula, which ultimately form tumors in the brain). These microinjection sites may be optimal since they recapitulate where solid tumors are found in clinical settings.

This review also addresses the relevance of the hours-post-fertilization period of the zebrafish for microinjection. Injections at the blastula stage, which are undifferentiated, demonstrate the remarkable tropism of the brain tumor cells from patients: they finally form microtumors in the brain and other regions of the central nervous system of the zebrafish [88], showing that the human cells migrate and thrive in the specific microenvironment of the brain. Finally, the number of cells is a crucial factor in proliferation assays. Not only due to the fact that it is essential to determine the baseline to evaluate proliferation, but also since the higher the number of cells, the higher the probability that it will alter the normal development of the zebrafish and cause false conclusions (higher mortality of the transplant not due to the malignancy of the cells, but due to nutrient depletion). All of these factors are essential to consider in every assay, but they can limit the comparisons between experiments in primary xenotransplants and cell lines.

### 4.3. Model Limitations

There are differences between zebrafish and humans that need to be considered [52]. As mentioned previously, the optimal temperature for zebrafish (28 °C) and human cells (37 °C) is different. However, studies using xenotransplantation of tumor cells into the zebrafish model have successfully moved towards clinical studies in personalized medicine [97]. In their most recent review (2019), Da Hora et al. highlights that it is challenging to study GB pathology in the neuro-oncology field [121]. They state that using cell lines as a model is not the best option since it does not represent the phenotype and the genotypic mutations of glioma cancer stem-like cells (GSCs). For that reason, scientists are turning more towards patient-derived cells xenotransplants as a more reliable model [121].

There are other limitations to using zebrafish in xenotransplantation experiments, arising from the phylogenetic distance between teleost fish and mammals. For example, orthopedic implantation is impossible due to the lack of corresponding organs in the model [37,52]. Furthermore, myelinated axonal sheaths do not develop in the zebrafish until four to seven days post-fertilization [122], affecting the invasion of implanted glioma cells [123]. Besides, in zebrafish embryos, the blood-brain barrier (BBB) does not develop up to 3 dpf [124] and is not mature for another seven days [125]. This observation is crucial for testing drugs targeting glioma cells since not all compounds may cross the BBB, and the developmental stage is essential for the experimental observation of pharmacologic effects in the model and clinical settings [52].

### 4.4. Study Limitations

We acknowledge limitations to our study, including selection bias due to our strict inclusion and exclusion criteria. We mitigated it by searching multiple sources in distinct databases (doctoral thesis, gray literature, published articles) across different research areas. Nonetheless, only five published articles were eligible for this systematic review, limiting the potential analysis of the different methodologies used by various authors. However, this review can conclude on essential parameters for developing the primary xenotransplant and encourages researchers to potentially standardize a protocol to develop a reliable and reproducible transplant.

## 5. Conclusions

Based on these findings, the potential of GB isolation and culture cells makes it a valuable method that mirrors the molecular characteristics of the original tumor, with economic, ethical, and experimental advantages compared to xenotransplant in other animal models. Primary xenotransplant of CNS neoplasms in zebrafish remains a growing area of research. The high heterogeneity in the protocols and difficulty in culturing most patient-derived tumors is a challenge to overcome, and the number of studies will surely increase in the upcoming years. Patient-derived CNS tumor cells xenotransplants into zebrafish rise as a valuable platform that can guide clinical treatments in a personalized way, with the ultimate goal of improving the outcomes for patients living with GB and NB and their complications.

## Figures and Tables

**Figure 1 cells-11-01204-f001:**
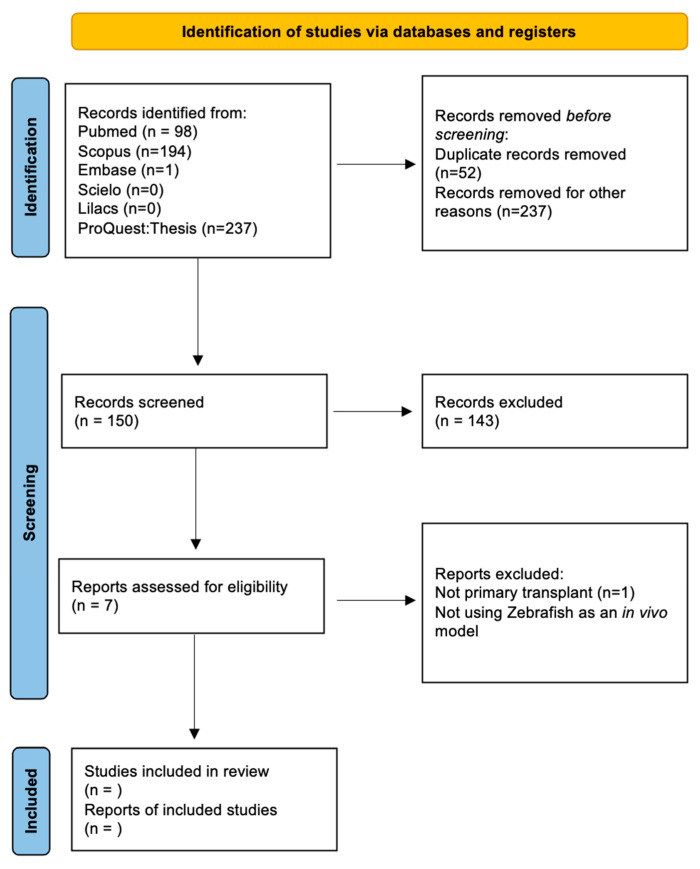
Scheme showing the systematic review process based on the PRISMA 2020 statement [85].

**Figure 2 cells-11-01204-f002:**
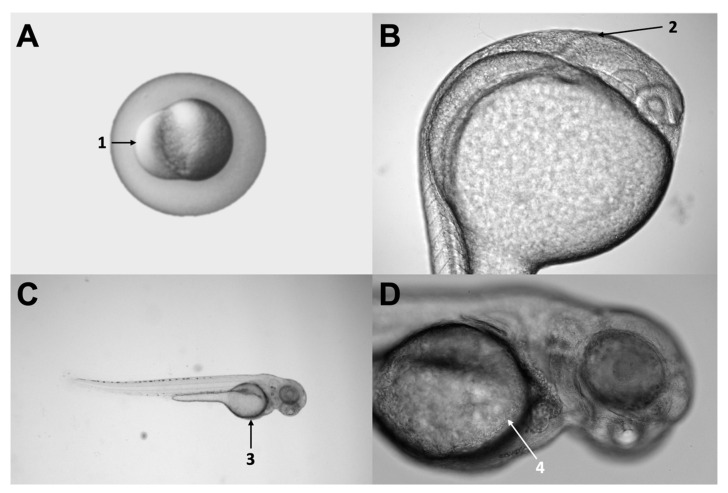
Different injection sites used in the articles evaluated. Panel (**A**) shows a three hour-post-fertilization (hpf) zebrafish embryo, indicating the injection site (number 1) used by Pudelko [88]. Panel (**B**) shows a 24 hpf larva, indicating the midbrain-hindbrain boundary (number 2), a relevant anatomical landmark, and a common injection site used at a different stage by Rampazzo, Wenger, and Banasavadi-Siddegowd (7 pf, 2 pf, and 36 hpf, respectively) [86,87,89]. Panel (**C**) shows a 48 hpf larva, indicating the yolk sac (number 3), which was the injection site used by Wrobel [90]. Finally, panel (**D**) shows a zoom-in of the same larvae, denoting the yolk sac and the duct of Cuvier in more detail (number 4). Original pictures were taken by the authors of this review.

**Figure 3 cells-11-01204-f003:**
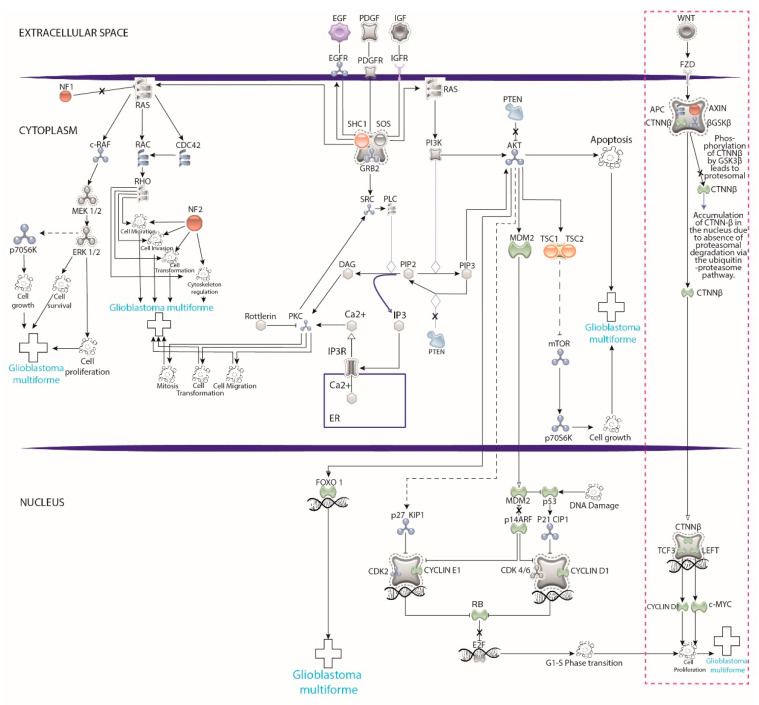
Pathways involved in the development and establishment glioblastoma (GB) at the molecular level. The pink dotted rectangle highlights that research using xenotransplants into zebrafish has focused only on the Wnt pathway, leaving a significant opportunity window to explore the remaining pathways. Graphics created in illustrator program based on the model of https://geneglobe.qiagen.com (accessed on 10 July 2021).

**Table 1 cells-11-01204-t001:** Summary of articles with patient-derived xenotransplant of CNS neoplasms in zebrafish.

Tumor Type	Characteristics of the Model Used	Microinjection Site	Molecules Screened	Tumor progression Parameters: Proliferation (P) Angiogenesis (ANG) Metastasis (Met) Apoptosis (Ap)	Reference
	Zebrafish embryo (wildtype AB strain)	Ventricles (2 dpf)	Temozolomide Etoposide and Vincristine (in-vitro)	P (in-vitro) ANG (No) Met (No) Ap (No)	[86]
	Wild-type or transgenic zebrafish larvae	Midbrain-hindbrain boundary (7 dpf)	None	P (in vitro and in vivo) ANG (in vitro) Met (No) Ap (No)	[87]
GB	Zebrafish embryos (Blastula)	Injection at the blastula stage of the zebrafish	Temozolomide, tyrosine kinase inhibitors (TKI), erlotinib (in vivo).	P (in vivo) ANG (in vivo) Met (No) Ap (No)	[88]
	Transparent casper mutant zebrafish (roy;nacre)	Midbrain-hindbrain boundary (36 hpf)	PRMT5 Inhibitors (in vivo)	P (in-vitro) ANG (No) Met (No) Ap (in vitro)	[89]
NB	Zebrafish Lines TE4/6 wildtype strain an Tg(*fli1:EGFP*)	Yolk sac of larvae (48 hpf)	Doxorubicin, vorinostat, panobinostat, tubastatin A (in vivo)	P (in vitro- in vivo) ANG (No) Met (in vivo) Ap (indirectly in vivo)	[90]

**Table 2 cells-11-01204-t002:** Comparison of the main features of xenotransplants in zebrafish larvae and mice.

	Zebrafish PDx in Larvae	Mouse PDx
Assay Duration	5–7 days	Weeks to months
Transparency allowing assessment via microscopy	Yes	No
Latency to tumor formation	Short	Long
Drug screening throughput	High	Low
Pharmacokinetic and dose optimization	No	Yes
Number of cells per recipient	10^2^	10^5^–10^7^
Cost	Low	High

PDx, patient-derived xenograft. Table created based on the review: Zebrafish patient avatars in cancer biology and precision cancer therapy [97].

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
