# Peer review of "Patient-Derived Xenotransplant of CNS Neoplasms in Zebrafish: A Systematic Review"

_cells, 2022, doi:10.3390/cells11071204_

Round 1
Reviewer 1 Report
In this systemic review, Sarmiento et al compare 6 studies modelling CNS neoplasms using the xenotransplant of primary tumor cells to the developing zebrafish. Authors provide detailed comparison of methodology and outcomes of these carefully selected studies and discuss their implementation in the understanding of the neoplasms. Overall, it is very structured and well written review with carefully described criteria for literature selection. This reviewer also finds the review very informative and highly relevant. In order, to increase the completeness of the review, this reviewer would suggest a shorth paragraph on the xenotransplants in the mammalian system. This would be beautiful for the comparison how the zebrafish system stands compared to the mammalian systems and would also probably allow authors to make their point about the relevance of the zebrafish system even more.
In addition, this reviewer finds it difficult to follow if the data discussed come from the 6 selected studies or overall neoplasms literature. Authors should outline this in the text. It is only summarized in the table and not easy to follow throughout the text.
Finally, in the model limitations, authors rather discuss the general limitations of the xenografts of primary cells. They should at least add concerns that would be zebrafish specific if any.
Reviewer 2 Report
This is very good paper, which comprehensively discuss zebrafish NB and GBM xenograft models. Authors discussed both prons ans cons of this models, methodological issues, as well as data obtained from those studies. It is definiately interesting topic for scientists.
Reviewer 3 Report
The manuscript entitled "patient-derived xenotransplantation of CNS neoplasms in zebrafish: a systematic review" describes the advantages of zebrafish embryos as avatars for xenotransplantation experiments and translational research. In my opinion, this review has no main message, except that (1) transplantation of human glioblastoma cells in zebrafish embryos is possible. (2) Thus far, only five works (maybe more?) have used primary cells in their studies. Perhaps, a reason why fewer works have been conducted on xenotransplantation of primary cells is that they are hard to culture??
Overall, I found the quality of this article very low. I got more impression that the authors intended to provide statistics of how many studies have been done in this field; the biological findings and background were underrepresented. Further, the manuscript contains many incorrect or contradictory statements; it is not well structured (repetitive sentences). I missed additional information or serious biological debate compared to other reviews on this topic. Hence, I can't entirely agree with the authors that this article "highlights valuable information" (line 411) and cannot support this manuscript to be published.
As examples, some of the major issues are listed below:
1- It is unclear why the authors added a Material and Methods section in a review article? This section, including Figure 1, describes the search for the literature, a normal procedure that every scientist does! Besides educative purposes for undergraduate students, this section is not informative for researchers.
2- It is unclear what the authors want to say in lines 78-80, and this statement is also contradictory to what they say in line 107. Overall, it appears that the definition of "cancer in vivo model" and "xenotransplantation model" were mixed in this manuscript.
3- Line 88: "hundreds of animals can be maintained on a multi-well plate..." I guess the authors mean embryos/larvae?
4- The statement that "almost all genes in zebrafish are duplicated" (line 95) is incorrect. Indeed, zebrafish has a complex genome, but many gene duplications are due to the third whole-genome duplication event in the teleost ancestor. Besides that, the genome complexity is rather relevant for modeling human diseases in this species and less an issue for assessing the proliferation/apoptosis of xenotransplanted human cells in avatars. Also, the lack of antibodies for zebrafish proteins is not a limitation for xenotransplantation assays. It is not clear what kind of proteomic analysis the authors mean in the context of the PDX model.
5- Line 109: What do the authors mean by "zebrafish models have intrinsic features"? Also, the statement "xenotransplantation can be performed in several hundred adult zebrafish and a few thousand zebrafish larvae in a single day by a single operator" (line 112) is possible in theory but unrealistic in practice (based on my >15 years of experience in this field).
6- Section 3.2. is also one of the examples that the authors described only the technical part of different works (injection of cells into different places) but did not evaluate or compare them. So, what is the conclusion? Is it not better to inject glioblastoma cells directly into the brain, where their natural environment is, or the place of injection is irrelevant?
7- The authors described in Table 1 the zebrafish strains in those studies have used them for transplantation experiments. Now, the question arises of whether there is a difference in zebrafish strains that need to be considered for the efficiency of xenotransplantation? Why is this information important?
8- Mutants/transgenic fish are not strains! The authors must correct the sentence in line 301.
9- Figure 3 is dispensable. In particular, the authors mentioned that "most of the molecules involved here have not studied..." Further, the texts in the figure are not readable, and again, the information value of this figure is unclear. How this figure helps me to understand the manuscript?
10- Line 349: Implantation? should not be "transplantation"?
Round 2
Reviewer 1 Report
Authors addressed all concerns raised.